# Decrease of Greenhouse Gases during an In Vitro Ruminal Digestibility Test of Forage (*Festuca arundinacea*) Conditioned with Selenium Nanoparticles

**DOI:** 10.3390/nano12213823

**Published:** 2022-10-29

**Authors:** Uriel González-Lemus, Gabriela Medina-Pérez, Armando Peláez-Acero, Rafael Germán Campos-Montiel

**Affiliations:** Instituto de Ciencias Agropecuarias, Universidad Autónoma del Estado de Hidalgo, Av. Rancho Universitario s/n Km. 1., Tulancingo C.P. 43600, Hidalgo, Mexico

**Keywords:** nanoparticles, selenium, digestibility, bioactive compounds, methane

## Abstract

The *Festuca arundinacea* Schreb. is one of the most used forage grasses due to its duration, productivity, great ecological breadth, and adaptability. Livestock has been criticized for its large production of greenhouse gases (GHG) due to forage. The advancement of science has led to an increase in the number of studies based on nanotechnologies; NPs supplementation in animal nutrition has found positive results in the fermentation of organic matter and the production of fatty acids and ruminal microorganisms. The objectives of this study were (1) to evaluate the in vitro digestibility of forage containing selenium (Se) nanoparticles (NPs), and to identify the specific behavior of the ruminal fermentation parameters of *F. arundinacea* Schreb. and (2) quantify the production of greenhouse gases (total gas and methane) (3) as well as the release of bioactive compounds (phenols, flavonoids, tannins, and selenium) after fermentation. Three treatments of SeNPs were established (0, 1.5, 3.0, and 4.5 ppm). The effects of foliar fertilization with SeNPs son digestion parameters were registered, such as the in vitro digestion of dry matter (IVDM); total gas production (*A*_total gas_) and methane production (*A*_CH_4__); pH; incubation time(to); the substrate digestion rate (S); t_Smax_ and the lag phase (L); as well as the production of volatile fatty acids (VFA), total phenols, total flavonoids, and tannins in ruminal fluid. The best results were obtained in the treatment with the foliar application of 4.5 ppm of SeNPs; IVDMD (60.46, 59.2, and 59.42%), lower total gas production (148.37, 135.22, and 141.93 mL g DM^−1^), and CH_4_ (53.42, 52.65, and 53.73 mL g DM^−1^), as well as a higher concentration of total VFA (31.01, 31.26, and 31.24 mmol L^−1^). The best results were obtained in the treatment with the foliar application of 4.5 ppm of SeNPs in the three different harvests; concerning IVDMD (60.46, 59.2, and 59.42%), lower total gas production (148.37, 135.22, and 141.93 mL g DM^−1^), and CH_4_ (53.42, 52.65, and 53.73 mL g DM^−1^), as well as a higher concentration of total VFA (31.01, 31.26, and 31.24 mmol L^−1^). The *F. arundinacea* Schreb. plants fertilized with 4.5 ppm released—in the ruminal fluid during in vitro fermentation—the following contents: total phenols (98.77, 99.31, and 99.08 mgEAG/100 mL), flavonoids (34.96, 35.44, and 34.96 mgQE/100 g DM), tannins (27.22, 27.35, and 27.99 mgEC/100g mL), and selenium (0.0811, 0.0814, and 0.0812 ppm).

## 1. Introduction

Greenhouse gas emissions (GHG), carbon dioxide (CO_2_), and methane (CH_4_) are compounds that directly affect climate change because these molecules can absorb radiant energy that reaches the Earth [1]. Agriculture and livestock production significantly contributes to anthropogenic greenhouse gas (GHG) emissions into the atmosphere [2].

The primary anthropogenic sources of GHG come from the agricultural sector since ruminants emit 100 million tons of CH_4_, representing approximately 20% of the total annual emissions [3]. CH_4_ is one of the three main greenhouse gases, together with CO_2_. CH_4_ emissions are the second most important anthropogenic GHG in global warming [4]. The CH_4_ production in ruminants is influenced by enteric fermentation, depending on the diet(grains and forages), the digestibility rate of the feed, previous processing, and the frequency of feeding. It represents a loss and, in environmental terms, favors the degradation of the ozone layer and global warming [5,6]. Different strategies have been developed to decrease CH_4_ production in ruminants, such as using vaccines, genetic selection, animal nutrition, and chemical substances [7,8,9]. Some alternatives include bio-fortified crops with selenium, such as *Brachiaria spp*, which has been shown to influence ruminal fermentation by reducing the production of total gas and CH_4_ [10,11].

Selenium nanoparticles (SeNPs) also have a wide range of potential advantages: antimicrobial agents, growth promoters, and nutraceuticals. In addition, SeNPs shows less toxicity in plants than the use of other inorganic forms of Se [12,13].

González-Lemus et al. [14] demonstrated that forages can be bio-fortified with a foliar application of Se nanoparticles, resulting in an increase in this element in forage plants and an improvement in nutritional characteristics. In recent years, the health benefits of Se for farm animals have been demonstrated, improving reproductive performance and immune function, among others [15]. In addition, the ruminal proportion of volatile fatty acids (VFA) can be increased; these effects may vary with the concentration in ruminant diets [16,17,18]. Forages are the main supporting feature of sustainable agriculture and constitute one of the primary resources for the success of the animal production system. *F. arundinacea* Schreb. is a cool-season perennial grass grown for pasture and silage. The objectives of this study were: (1) to evaluate the in vitro digestibility of a bio-fortified forage conditioned with SeNPs, (2) to measure the ruminal fermentation parameters of *F. arundinacea* Schreb., (3) and to quantify the production of greenhouse gases (total gas and methane) as well as the release of bioactive compounds (phenols, flavonoids, tannins, and selenium) after fermentation. 

## 2. Materials and Methods

### 2.1. Sample Collection and Preparation

#### 2.1.1. Production of Festuca arundinacea Schreb. Grass with Selenium

This study was carried out in a greenhouse at the Instituto de Ciencias Agropecuarias, Universidad Autonoma del Estado de Hidalgo (UAEH), located at coordinates 20°3′36.44″ N and 98°22′53.26″ W of the meridian of Greenwich. Certified grass seed (*Festuca arundinacea* Schreb.) with a purity of 99.0% and a germination capacity of 99.0% was obtained from “El Instituto Nacional de Investigaciones Forestales, Agrícolas y Pecuarias” (INIFAP), in Mexico. Sowing was conducted on this forage, with a manual rounder, with a density of 40 kg/ha [19]. According to FAO/UNESCO soil classification system, the soil was a haplic *phaeozem* with pH 7.54 and electrolytic conductivity of 5.3 dS m^−1^, a water holding capacity (WHC) of 625.01 g kg^−1^, an organic carbon content of 3.6 g C kg^−1^ soil, and a total inorganic N content of 0.21 g N kg^−1^ soil. The temperature during the experiment was 25–28 °C during the day and 16-19 °C at night from May to September 2020. Selenium bio fortification of the forage was carried out by foliar application with SeNPs at different concentrations (1.5, 3.0, and 4.5 ppm) in an experimental design with three treatments and a control, all with five replicates. The SeNPs with a size < 50 nm were purchased from Materiales Nanoestructurados S.A de CV (San Luis Potosí, México).

The foliar application of Se NPs was conducted 21 days after seed germination, during the vegetative phase of the crop, following the methodology proposed by [20] with some modifications; only distilled water was applied to the control. Three harvests wera period of 90 days each.

#### 2.1.2. Nanomaterials

Nanoparticles applied in the experiment were purchased from Materiales Nanoestructurados S.A de CV (San Luis Potosí, México). The physicochemical characteristics of selenium nanoparticles are listed in Table 1.

#### 2.1.3. Sample Conditioning

The forage grass (*Festuca arundinacea* Schreb.) fertilized by foliar route with the different concentrations of Se from the three different harvests was dehydrated in a drying oven with airflow at a maximum temperature of 60 °C for a period of 72 h. Subsequently, the treatments were ground in a turbine pulverizer mill with a 2 mm sieve and stored in totally closed containers until analysis

### 2.2. Obtaining the Ruminal Fluid

The fresh ruminal fluid was obtained from two sheep (Hampshire breed, 56 ± 1.4 kg live weight), provided with a ruminal cannula, and fed with stubble for 15 days before sampling. The ruminal liquid was filtered through filter paper to remove macroparticles of organic matter and stored in a flask with a continuous flow of CO_2_ to maintain anaerobic conditions. The experiment followed the protocol of the Institutional Bioethics Committee for managing and using laboratory animals of the UAEH (under the guidelines of the Law for the protection and dignified management of animals in the state of Hidalgo based on official standards) [21,22].

### 2.3. In Vitro Incubation 

For fermentation and in vitro gas production kinetics, the methodology described by [23], with some modifications, was used. Five repetitions per treatment were established. In 120 mL serological vials, 0.5 g of a dry and ground substrate and 50 mL of culture medium (biodigester) were added to set the assay. According to [24], the culture was composed of a 1:10 solution of fresh ruminal fluid in a reduced-mineral buffer (150 mL of mineral solution I (6 g K_2_HPO_4_ (Merck^®^) in 1000 mL of distilled H_2_O) and 150 mL of mineral solution II (6 g K_2_HPO_4_ (Meyer^®^) + 6 g (NH_4_)2SO_4_ (Merck^®^) + 12 g NaCl (Meyer^®^) + 2.45 g MgSO_4_ (Sigma^®^) + 1.6 g CaCl-2H_2_O (Meyer^®^) in 1000 mL of distilled H_2_O); 100 mL of 8% Na_2_CO_3_ solution (Merck^®^ ); 100 mL of reducing solution (12.5 g L-cysteine (Sigma^®^) + 12.5 g Na_2_S-9H_2_O (Meyer^®^) + 2 mL of a 2N NaOH solution (Meyer^®^) in 100 mL of distilled H_2_O, which was heated to 250 °C and subsequently cooled, and 0.1 mL of 0.1% resazurin (Sigma-Aldrich^®^)) was added. The biodigesters were subjected to a constant flow of CO_2_ to maintain anaerobic conditions. Subsequently, they were hermetically sealed (manual crimping machine, Wheaton, IL, USA) utilizing a silicone stopper and an aluminum capsule with a removable center. The flasks were incubated in a water bath at 39 °C. 

### 2.4. Total Gas and Methane Estimation

Water volume displacement was measured in vitro gas production by inserting a hypodermic needle attached to a graduated glass water column through the silicone plug. Measurements were made at 0, 2, 3, 5, 7, 8, 9, 10, 16, 20, 24, 28, 32, 44, 56, 68, 80, and 92 h of incubation (5 independent replicates per substrate). The gas produced was used to obtain the gas production kinetics parameters and gas production rate (S, h^−1^). The production and kinetics of CH_4_ were carried out following the methodology proposed by [25]. The biodigester was attached to a hermetically sealed trap vial filled with a NaOH (2N) solution and coupled to a Taygon^®^ hose (2.38 mm internal Ø and 45 cm long) with hypodermic needles (20 G × 32 mm) at the ends. The production of CH_4_ was measured at 0, 8, 16, 24, 48, 56, and 72 h (5 repetitions per substrate). The volume change was measured as the displaced mL of the NaOH (2N) solution since CO_2_ reacts with NaOH forming Na_2_CO_3_ [26].

### 2.5. Determination of Volatile Fatty Acids by Gas Chromatography

VFA was determined by following the methodology proposed by [27] with slight modifications. After fermentation (96 h), 2 mL of supernatant was taken from the content of each biodigester that contained the different substrates of the three harvests. It was centrifuged at 12,000× *g* for 20 min. Then, 1 mL of the supernatant was taken and added to a tube containing 0.5 mL of 25% metaphosphoric acid. Subsequently, the mixture was left to stand for 60 min before centrifugation at 5000× *g* and stored at −20 °C in 2 mL vials for later determination of VFA. Samples were analyzed using a Perkin-Elmer XL gas chromatograph (Waltham, MA, USA) equipped with an FID. Nitrogen was used as carrier gas with a purity of 99.99% and a 30 m × 0.25 mm column. The initial oven temperature was 70 °C for 3 min, then increased 15 °C/min to 200 °C. The final temperature was maintained for 3 min. The injector temperature was programmed at 250 °C. For the determination of VFA, the retention times of acetic, propionic, and butyric acid were determined at 3.2, 4.5, and 5.6 min, respectively.

### 2.6. In Vitro Dry Matter Digestibility

After fermentation and the incubation period of the different experimental substrates (92 h), the calculation of the in vitro digestibility of dry matter (IVDMD) of the different treatments of the three pasture harvests was estimated following the methodology proposed by [28]. The residues of each fermentation were collected by filtration through a Buchner funnel (filter paper F, fast MOD.617, PV NO.1034). The IVDMD was estimated by drying the residual material at 60 °C for 72 h and calculated as the difference between the initial and residual dry matter weight.

The mathematical description of these profiles helps compare the characteristics of the substrates or the fermentation environment. The adjustment of the experimental data to the Logistic model using the Sigma Plot 12© software (SPSS Inc.; Chicago, IL, USA) allowed obtaining Equation (1); where *y* (mL g^−1^ DM) denotes the amount of accumulated gas produced per gram of dry matter (DM) at time *t* (h) during incubation. An (ml g^−1^ DM) represents the maximum gas production at an infinite time. To (h) is the incubation time in which half of *A* has been produced, and *b* is a dimensionless constant that determines the characteristic profile and, therefore, the curve’s inflection point.
(1)$$y=\frac{A}{1+{\left(\frac{t0}{t}\right)}^{b}}$$

The indicator inflection points of the delay phase (*L*) or Equation (2) resulting from *dy*/*dt* are:(2)L=t0[b−1b+1]1b

Considering the disappearance of the substrate (*P*), as a first-order kinetics, it is the rate of digestion of the substrate (*S*). Equation (3) for values of *b* > 1 increases until it reaches a maximum (*Smax*) when the size of the microbial population already does not limit the fermentation of the food. The time when *Smax* is reached; is given by the resolution of *dS/dt* = 0. Equation (4) [29].
(3)$$S=\frac{1}{P}\frac{dP}{dt}=\frac{bt^{b-1}}{to^{b}+t^{b}}$$
(4)$$t_{Smax}=to{\left(b-1\right)}^{1/b}$$

### 2.7. Determination of Bioactive Compounds after In Vitro Digestion

#### 2.7.1. Determination of Total Phenols

The total phenol content was determined using the Folin–Ciocalteau method according to [30] with some modifications. After fermentation (96 h), 5 mL of the supernatant mixture was taken from the content of each biodigester from the different treatments (0, 1.5, 3.0, and 4.5 ppm). The mixture was centrifuged at 12,000× *g* for 20 min. In a test tube, 0.5 mL of the supernatant and 2.5 mL of 10% Folin–Ciocalteau reagent (v/v) were added and incubated in the dark for 8 min, and 2.0 mL of Na_2_CO_3_ 7.5% (p/v) was added. After the mixture was vortexed (Vortex WM-10), it was allowed to react for 60 min in the dark at 22 °C. Finally, the absorbance at 765 nm was measured with a spectrophotometer (Jenway 6715 (Staffordshire, UK) using distilled water as a blank. A calibration curve was developed using gallic acid as a standard (0–100 mg L^−1^), and the content of total phenols was expressed as equivalent mg of gallic acid per 100 mL of ruminal fluid (mg GAE/100 mL).

#### 2.7.2. Total Flavonoids

The determination of flavonoids was carried out according to [31], with some modifications. After fermentation (96 h), 5 mL of supernatant was taken from the content of each biodigester from the different treatments (0, 1.5, 3.0, and 4.5 ppm). The mixture was centrifuged at 12,000× *g* for 20 min. Then, 0.5 mL of the supernatant was used with 0.5 mL of a 10% (p/v) aluminum trichloride (AlCl_3_) solution and 0.5 mL of 0.1 mM sodium nitrate. Subsequently, the mixture was vortexed and incubated for 60 min at 22 °C. Then, 250 µL of NaOH (1 M) was added to stop the reaction. After incubation, the absorbance of the mixture was measured at a wavelength of 415 nm using a UV-VIS spectrophotometer (Jenway 6715, Staffordshire, UK). A calibration curve was developed using quercetin as a standard (0–100 mg L^−1^) and methanol as blank; the results were expressed in mg equivalents of quercetin per 100 mL of ruminal fluid (mg QE/100 mL).

#### 2.7.3. Total Tannins

The tannin content was measured according to [32] with some modifications. A 0.1 M FeCl_3_ solution was made using 0.1 M HCl as solvent. After fermentation (96 h), 5 mL of supernatant was taken from the content of each biodigester from the different treatments (0, 1.5, 3.0, and 4.5 ppm). The mixture was centrifuged at 12,000× *g* for 20 min. Then, 600 µL of 0.1M FeCl_3_ was added to a test tube, 200 µL of the supernatant extract was left for 5 min in the dark, and the mixture was allowed to react for 10 minutes at 22 °C. The samples were read with a UV-VIS spectrophotometer (Jenway 6715, Staffordshire, ST15 OSA, UK) at 720 nm. A treatment was prepared with the same conditions by replacing the supernatant with an ethanol/water solution (1:1), which was analyzed. The result was subtracted from the readings of the different treatments. A catechin standard curve was developed, and the results were expressed as mg equivalents of catechin per 100 mL of ruminal fluid (mg EC/100 mL).

#### 2.7.4. Selenium Determination in Ruminal Fluid by Atomic Absorption Spectrophotometer

The analysis of Se content in the grass (*Festuca arundinacea* Schreb.) was carried out using the acid digestion procedure, pre-reduction of Se, and detection by hydride generation atomic absorption spectrometry (HG-AAS), following the methodology of [33] with some modifications. A total of 10 mL of ruminal fluid, ground, was placed in a digestion tube, 20 mL of concentrated HNO_3_ was added, and the mixture was heated at 175 °C for 60 min. The temperature of the heated sample decreased to 150 °C, and it was incubated for 90 min. Five milliliters of H_2_SO_4_/HNO_3_ (2:1) was added to the tube, heated to 175 °C for 60 min, cooled to 20 °C, and then 2 mL of H_2_O_2_ was added dropwise and then heated to 140 °C for 10 minutes. The digestion result (whitish or slightly yellow) was diluted to 25 mL with double-distilled water. Subsequently, a pre-reduction of Se^6^ to Se^4^ was carried out, 5 mL of 6 M HCl was added to 5 mL of the previous mixture, and it was heated at 60 °C for 120 min. In all cases, the standard addition method was used. Equipment was set with a Se hollow cathode lamp with a spectral slit width of 2.0 nm and an absorption wavelength of 196 nm with a hydride generator (Perkin Elmer MHS-15, (Shelton, CT, USA), consisting of a peristaltic pump and injection valve. The acetylene gas flow was 2.5 L min^−1^, and the airflow was 11.5 L min^−1^. The results were expressed as ppm of Se per L.

### 2.8. Statistical Analysis

A completely randomized design with five repetitions was used for the statistical analysis. The statistical model was: Yij = μ + τi + εij; where Yij is the response variables in repetition j of sample i; μ is the overall mean; τi is the effect of the i-th type of substrate, and εij is the random error of the i-th type of sample in the j-th iteration. The results were analyzed with an analysis of variance. When significant differences (*p* < 0.05) were observed between the treatments, the Tukey test was applied (SigmaPlot 12.0 software program Systat Software Inc., London, UK).

## 3. Results

### 3.1. Digestibility Parameters

The pH values of the ruminal liquid ranged from 6.52 to 6.66 for the treatments (0, 1.5, 3.0, and 4.5 ppm) and did not show significant differences (*p* > 0.05) (Table 2). In this study, the selenium foliar-enriched treatment with 4.5 ppm presented a higher IVDMD (60.45, 59.24, and 59.42%) in the three established harvests compared to the other treatments. There was a significant difference (*p* < 0.05) between kinetic profiles; the latency time (L) was lower (0.47, 0.45, and 0.61) in the selenium foliar-enriched treatment with 4.5 ppm enriched with 4.5 ppm, and it presented a higher time of latency (0.85, 1.13, and 1.08). The concentration of Se was higher in forage, with 4.5 ppm of SeNPs. The *Smax* parameter showed significant differences (*p* < 0.05 (Table 2)).

### 3.2. In Vitro Gas and Methane Production

The total accumulated gas production of the treatments in the different harvests shows a sigmoidal behavior during the 92 h fermentation. The control treatment generated the highest volume of greenhouse gases, compared to the other treatments, foliar selenium enriched with SeNPs, showing a similar behavior in each harvest. On the other hand, the foliar-enriched forage treatment with 4.5 ppm of SeNPs presented the lowest in vitro accumulated gas production in the three crops (148.37, 135.22, and 141.93 mL g^MS−1^) (Figure 1) compared to the other treatments., with significant differences (*p* < 0.05), which showed similar behavior in each harvest. The CH_4_ production profiles showed that the produced volume tends to decrease in response to the foliar incorporation of Se NPs (Figure 2). The values presented by the control treatment were 7.88, 56.54, and 57.54 mL g^MS−1^ for each harvest that produced the highest volume of CH_4_. On the other hand, the treatment with 4.5 mg of Se NPs presented the lowest volume of CH_4_ in the three harvests (53.42, 52.65, and 53.73 mL g^MS−1^), with significant differences (*p* < 0.05) compared to the other treatments and in each of the harvests (Figure 2).

### 3.3. Determination of Bioactive Compounds in the Ruminal Fluid after Digestion

The concentration of total phenols, flavonoids, tannins, and selenium in the ruminal fluid after the digestion of the pasture treatments for forage enriched with Se NPs was determined. The results show a tendency to increase in response to the foliar incorporation of Se NPs in the grass. The 4.5 ppm treatment in the three harvests showed the highest concentration in terms of bioactive compounds’ total phenols (98.77, 99.31, and 99.08 mgEAG/100 mL), flavonoids (34.96, 35.44, and 34.96 mgQE/100 g DM), tannins (27.22, 27.35, and 27.99 mgEC/100g mL), and selenium (0.0811, 0.0814, and 0.0812 ppm), with significant differences (*p* <0.05) compared to the other treatments and in each of the harvests (Table 3).

### 3.4. Quantification of Volatile Fatty Acids (VFA)

The production of acetic and propionic acid in the foliar-enriched treatment with 4.5 ppm NPs at the end of fermentation showed significant differences (*p* < 0.05) between the treatments in each of the harvests (Table 3). Similarly, the highest concentration of total AGV (31.01, 31.26, and 31.24 mmol L^−1^) produced a decrease in acetate production and an increase in propionate, with significant differences (*p* < 0.05) between treatments and in each one of the harvests. On the other hand, the control treatment produced the highest content of acetic acid and the lowest concentration of total VFA (Table 4).

## 4. Discussion

The pH values of 6.0−6.7 improve the dry matter digestibility. The pH values did not decrease since the substrate for digestion does not favor the acidification of the ruminal fluid due to its chemical composition. The pH parameter in this study is similar to that reported by [34], where they obtained a pH of 6.5 when fermenting *Festuca arundinacea*. When pH values below 6.5 cause inhibition in the development of cellulolytic bacteria; CH_4_ production, which generates longer times in the delay phase (L); and a decrease in IVDMD [35,36]. 

In the IVDMD results, there were significant differences (*p* > 0.05) at the end of the fermentation (92 h). The foliar-enriched treatment with 4.5 ppm of SeNPs in the three harvests showed the highest digestibility of the matter (60.46, 59.24, and 59.42%). Evidence indicates that the supply of Se improves ruminal fermentation, digestibility of dry matter, organic matter, crude protein, and ethereal extract [37]. The authors of [38] used maralfalfa grass (*Cenchrus purpureus* Schumach.). They observed a degradability average of 48.9%. In contrast, in the present study, the degradability of *Festuca arundinacea* Schreb. ranged from 55.02% to 60.46%, which increased as SeNPs to the forage increased.

The decrease in total gas production can be attributed to increased phenolic compounds, flavonoids released in the ruminal fluid during fermentation, and foliar fertilization of SeNPs in the grass. Cajarville et al. [39] quantified the total gas production through an in vitro fermentation of 72 h of *Festuca arundinacea* forage, which produced 143.7 mL g DM^−1^, higher than the potential gas production of the fertilized grass with 4.5 SeNPs at 72 h of fermentation. 

The study carried out by [11] found that in the supply of Se (20, 40, 60, and 80 mg Se kg^−1^ DM) in oat hay, there was a decrease in the total gas production in the fermentation as the content of sodium selenite was higher. The incorporation of phenolic compounds such as tannins in a ruminal fermentation influence the microbiota due to their ability to bind to the microorganism’s cell wall, causing morphological changes or the secretion of extracellular enzymes that cause changes in their metabolism [37]. The author of [40] carried out an in vitro study where he demonstrated that incorporating polyphenols generates a decrease in CH_4_. Furthermore, a study by [11] showed that increasing the selenium content in the in vitro fermentation of oat hay decreases the production of greenhouse gases such as methane. On the other hand, [41] found that the supply of polyphenol-rich olive leaves as a dietary additive affects methane reduction during in vitro fermentation due to a decrease in the proportion of methanogenic archaea in the ruminal fluid. 

According to [42], selenium modifies the phenylpropanoid route in plant metabolism by increasing the enzymatic activity of phenylalanine ammonium lyase (PAL). Several authors found a positive correlation between the increased phenolic compounds in plants and the addition of Se. Similarly, Sharma et al. [43] attributed the increase in phenols and tannins in rice to the presence of Se. On the other hand, [44] showed that fortification with Se NPs (5 mg/L) in celery (*Apium graveolens* L.) increased total flavonoids up to 1.5 times more than that of the control group. Regarding the selenium concentration, a more significant amount of this element was found in the ruminal fluid when applying a forage conditioned with 4.5 ppm for SeNPs.

The changes in acetic acid and propionic acid could be attributed to the foliar enrichment of SeNPs, by influencing the rumen fermentation pattern, modifying the production of acetate to propionate, generating an increase in the concentration of propionic acid, directing ruminal fermentation towards a decrease in available H^+^, and thus decreasing CH_4_ production [18]. High production of total VFA is related to more efficient fermentation, which positively affects the availability of energy for the ruminant [38]. Similar studies prove the effect of Se in VFA modification, such as the work carried out by [42], where they evaluated the effect of Se-enriched yeast supplementation with an essential forage diet. The results show that the addition of Se, although it did not decrease the production of acetic acid, increased the concentration of propionic acid and the concentration of total VFA. Authors of [45,46] supplied massive sodium selenite in cows’ diets. They observed that the ratio of acetate to propionate decreased. 

## 5. Conclusions

Foliar fertilization of SeNPs increased the digestion rate and reduced the lag phase, which improves the digestibility of *F. arundinacea* Schreb. grass as a substrate. Treatments enriched with Se NPs tended to release a higher content of bioactive compounds (phenols, flavonoids, and tannins) and Se after in vitro digestion. Treatment with 4.5 ppm SeNPs produced a higher total VFA content, a lower total gas volume, and decreased methane gas content. The application of Se NPs could be an excellent alternative to be incorporated into forage crops that could help to improve digestibility parameters and reduce greenhouse gases.

## Figures and Tables

**Figure 1 nanomaterials-12-03823-f001:**
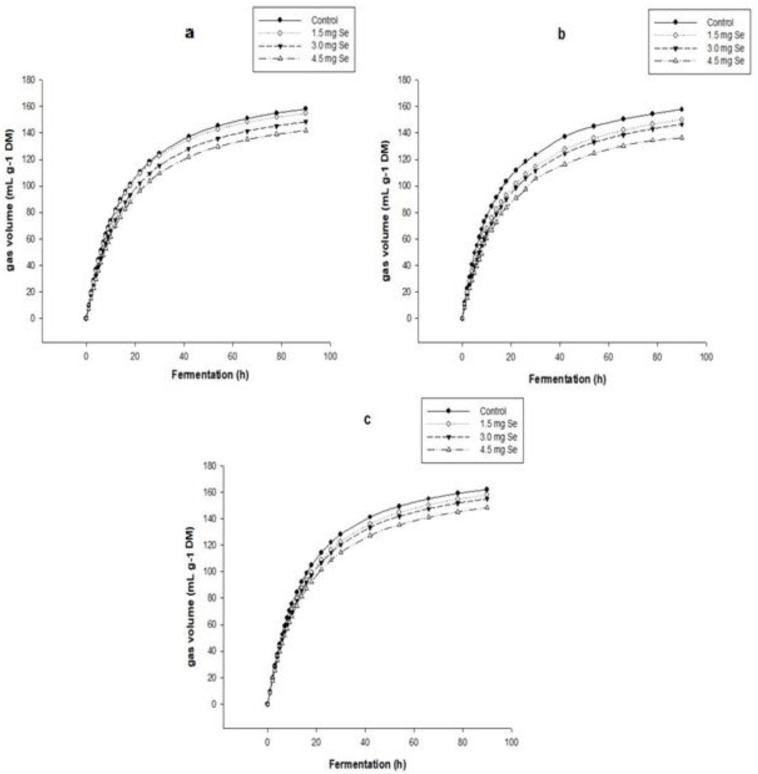
Cumulative total gas production profile during the 92 h fermentation in the different treatments and harvests. (**a**) First harvest; (**b**) Second harvest; (**c**) Third harvest.

**Figure 2 nanomaterials-12-03823-f002:**
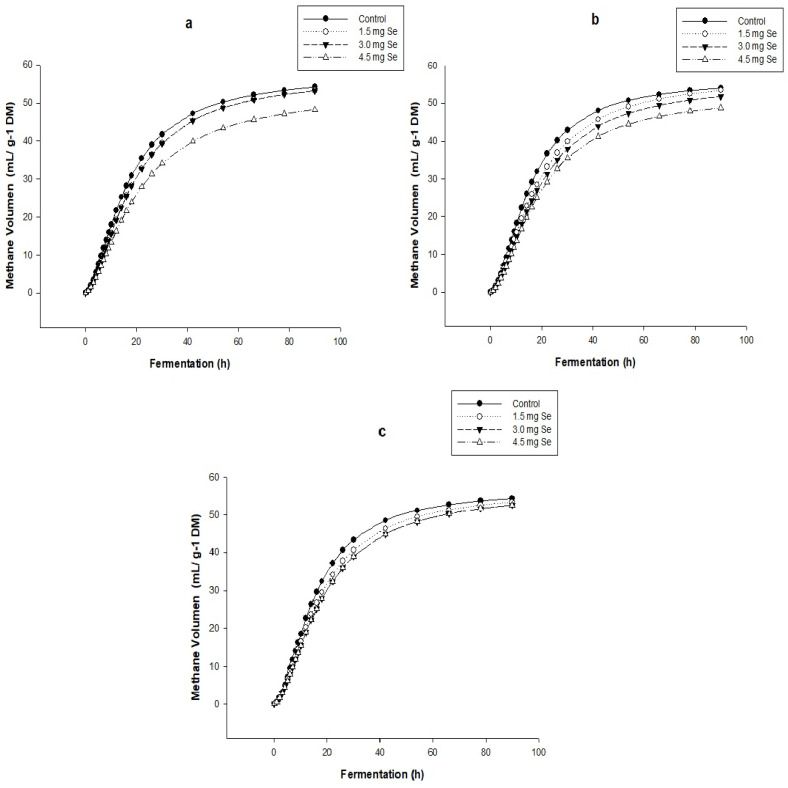
Accumulated methane production profile during the 92h fermentation in the different treatments and different harvests. (**a**) the first harvest; (**b**) the second harvest; (**c**) the third harvest.

**Table 1 nanomaterials-12-03823-t001:** Physicochemical characteristics of selenium nanoparticles used for foliar fertilizing the grass (*Festuca arundinacea* Schreb.) in a 90-day greenhouse experiment.

Attribute	SeNPs
Chemical formula	Se
Color	Gray
Density (g/cm^−3^)	4.81
Molecular weight	78.96
Melting point	960.8 °C
Boiling point	222.12 °C
Magnetic properties	Weakly ferromagnetic
Particle size Morphology	Less than 100 nm Spherical 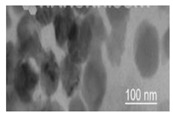

**Table 2 nanomaterials-12-03823-t002:** Digestibility parameters after 92 h of fermentation.

Selenium (ppm)	First Haverst	Second Haverst	Third Haverst
IVDMD (%)			
0	52.31 ± 1.35 ^cA^	50.54 ± 1.45 ^cA^	51.11 ± 1.15 ^cA^
1.5	55.83 ± 1.25 ^bA^	55.42 ± 1.38 ^bA^	55.02 ± 1.18 ^bA^
3.0	55.57 ± 1.12 ^bA^	55.77 ± 1.99 ^bA^	56.24 ± 1.11 ^bA^
4.5	60.46 ± 2.08 ^aA^	59.24 ± 1.75 ^aA^	59.42 ± 1.25 ^aA^
pH			
0	6.55 ± 0.55 ^aA^	6.52 ± 0.54 ^aA^	6.52 ± 0.78 ^aA^
1.5	6.52 ± 0.45 ^aA^	6.57 ± 0.63 ^aA^	6.66 ± 0.65 ^aA^
3.0	6.51 ± 0.44 ^aA^	6.65 ± 0.43 ^aA^	6.51 ± 0.54 ^aA^
4.5	6.54 ± 0.24 ^aA^	6.63 ± 0.45 ^aA^	6.64 ± 0.65 ^aA^
b (−)			
0	1.0922 ± 0.005 ^bA^	1.1089 ± 0.004 ^cA^	1.1073 ± 0.002 ^cA^
1.5	1.0458 ± 0.009 ^aA^	1.0834 ± 0.001 ^bA^	1.0973 ± 0.001 ^bA^
3	1.0557 ± 0.009 ^aA^	1.0793 ± 0.001 ^bA^	1.1071 ± 0.002 ^cB^
4.5	1.0562 ± 0.008 ^aA^	1.0503 ± 0.003 ^aA^	1.0716 ± 0.002 ^aB^
to (h)			
0	14.9089 ± 0.15 ^bC^	16.4926 ± 0.15 ^bB^	15.9474 ± 0.11 ^cB^
1.5	14.9428 ± 0.16 ^bB^	16.3713 ± 0.19 ^bA^	14.9305 ± 0.18 ^bB^
3	14.3876 ± 0.13 ^aB^	16.5697 ± 0.22 ^bA^	13.6736 ± 0.11 ^aC^
4.5	14.4339 ± 0.09 ^aB^	15.4877 ± 0.11 ^aA^	14.1273 ± 0.09 ^abB^
S (h^−1^)			
0	0.0505 ± 0.001 ^bA^	0.0444 ± 0.001 ^cB^	0.0461 ± 0.002 ^dB^
1.5	0.0559 ± 0.001 ^aA^	0.0468 ± 0.001 ^bC^	0.0500 ± 0.001 ^cB^
3	0.0566 ± 0.001 ^aA^	0.0465 ± 0.002 ^bC^	0.0537 ± 0.001 ^bB^
4.5	0.0564 ± 0.002 ^aA^	0.0533 ± 0.002 ^aB^	0.0555 ± 0.002 ^aA^
t_Smax_ (h)			
0	1.5547 ± 0.01 ^bC^	2.2329 ± 0.09 ^cA^	2.1243 ± 0.11 ^cB^
1.5	1.1708 ± 0.09 ^aB^	1.7821 ± 0.10 ^bA^	1.7861 ± 0.12 ^bA^
3	1.2230 ± 0.09 ^aC^	1.5829 ± 0.11 ^bB^	1.8177 ± 0.09 ^bA^
4.5	1.2094 ± 0.07 ^aA^	1.1954 ± 0.12 ^aB^	1.2063 ± 0.09 ^aA^
L (h)			
0	0.8551 ± 0.009 ^cC^	1.1393 ± 0.01 ^cA^	1.0836 ± 0.09 ^cB^
1.5	0.6950 ± 0.007 ^bC^	0.8395 ± 0.05 ^bB^	0.9094 ± 0.08 ^bA^
3	0.4715 ± 0.007 ^aC^	0.8033 ± 0.009 ^aB^	0.9271 ± 0.09 ^bA^
4.5	0.4777 ± 0.008 ^aB^	0.4537 ± 0.01 ^aB^	0.6113 ± 0.09 ^aA^

The results are expressed as means ± standard deviation. Capital letters ^ABC^ indicate significant differences (*p* < 0.05) in the samples (columns). Lowercase letters. ^a,b,c^. indicate significant differences (*p*<0.05) between the same samples in the three harvests.

**Table 3 nanomaterials-12-03823-t003:** The final concentration of total phenols, total flavonoids, total tannins, and Se after ruminal fermentation of *Festuca arundinacea* with Se NPs.

Se (ppm)	Ruminal Fluid before Digestion	Digestion First Harvest	Digestion Second Harvest	Digestion Third Harvest
Total phenols (mg/100 mL)				
0	4.95 ± 7.51 ^aB^	80.73 ± 1.26 ^dA^	81.91 ± 2.35 ^aA^	82.55 ± 2.91 ^aA^
1.5	4.93 ± 5.06 ^aB^	86.21 ± 2.52 ^cA^	85.41 ± 1.99 ^bA^	86.09 ± 1.75 ^bA^
3.0	4.98 ± 8.63 ^aC^	94.58 ± 1.72 ^bA^	91.99 ± 1.44 ^cB^	92.28 ± 2.12 ^cB^
4.5	4.96 ± 7.89 ^aB^	98.77 ± 1 43 ^aA^	99.31 ± 2.91 ^dA^	99.08 ± 1.66 ^dA^
Total flavonoids (mg/100 mL)				
0	1.37 ± 0.082 ^aB^	25.18 ± 0.70 ^dA^	23.99 ± 0.55 ^dA^	25.18 ± 0.47 ^dA^
1.5	1.41 ± 0.023 ^aB^	26.71 ± 0.23 ^cA^	25.19 ± 0.36 ^cB^	26.71 ± 0.94 ^cA^
3.0	1.38 ± 0.063 ^aB^	29.09 ± 0.63 ^bA^	28.18 ± 0.95 ^bA^	29.02 ± 0.86 ^bA^
4.5	1.39 ± 0.028 ^aB^	34.96 ± 0.74 ^aA^	35.44 ± 0.81 ^aA^	34.96 ± 0.48 ^aA^
Total tannins (mg/ 100 mL)				
0	1.96 ± 0.038 ^aB^	19.16 ± 1.55 ^cA^	17.16 ± 1.45 ^dA^	18.16 ± 1.38 ^cA^
1.5	1.98 ± 0.012 ^aB^	21.68 ± 1.23 ^cA^	21.44 ± 1.13 ^cA^	21.09 ± 1.37 ^cA^
3.0	1.97 ± 0.018 ^aB^	25.97 ± 0.94 ^bA^	25.24 ± 1.14 ^bA^	25.15 ± 0.64 ^bA^
4.5	1.96 ± 0.068 ^aB^	27.22 ± 1.05 ^aA^	27.35 ± 0.51 ^aA^	27.99 ± 0.45 ^aA^
Selenium (ppm)				
0	0.0008 ± 0.0005 ^aA^	0.0018 ± 0.0003 ^aA^	0.0013 ± 0.0004 ^aA^	0.0017 ± 0.0002 ^aA^
1.5	0.0005 ± 0.0002 ^bA^	0.0285 ± 0.0022 ^bA^	0.0254 ± 0.0014 ^bA^	0.0294 ± 0.0010 ^bA^
3.0	0.0004 ± 0.0005 ^cA^	0.0644 ± 0.0011 ^cA^	0.0611 ± 0.0016 ^cA^	0.0673 ± 0.0016 ^cA^
4.5	0.0006 ± 0.0002 ^dA^	0.0811 ± 0.0012 ^dA^	0.0814 ± 0.0018 ^dA^	0.0812 ± 0.0011 ^dA^

The results are expressed as means ± standard deviation. Capital letters ^ABC^ indicate significant differences (*p* < 0.05) in the different samples (columns). Lowercase letters ^a,b,c^ indicate significant differences (*p* < 0.05) between the same samples in different harvests.

**Table 4 nanomaterials-12-03823-t004:** Content of VFA after ruminal fermentation of *Festuca arundinacea* Schreb. whit SeNPs.

Selenium (ppm)	First Harvest	Second Harvest	Third Harvest
Acetic acid (mmol L^−1^)			
0	21.85 ± 0.35 ^bA^	21.97 ± 0.31 ^bA^	21.95 ± 0.15 ^bA^
1.5	21.71 ± 0.15 _bA_	21.93 ± 0.28 ^bA^	21.96 ± 0.18 ^bA^
3.0	21.49 ± 0.12 ^bA^	21.54 ± 0.09 ^bA^	21.67 ± 0.11 ^bA^
4.5	20.69 ± 0.08 ^aA^	20.68 ± 0.11 ^aA^	20.85 ± 0.15 ^aA^
Propionic acid (mmol L^−1^)			
0	5.45 ± 0.25 ^cA^	5.49 ± 0.24 ^cA^	5.42 ± 0.28 ^cA^
1.5	6.02 ± 0.35 ^cA^	5.77 ± 0.23 ^cA^	5.86 ± 0.25 ^cA^
3.0	7.11 ± 0.24 ^bA^	7.15 ± 0.13 ^bA^	7.51 ± 0.14 ^bB^
4.5	8.34 ± 0.14 ^aA^	8.63 ± 0.25 ^aA^	8.44 ± 0.15 ^aA^
Butyric acid (mmolL^−1^)			
0	1.54 ± 0.23 ^cA^	1.55 ± 0.12 ^cA^	1.51 ± 0.11 ^cA^
1.5	1.58 ± 0.19 ^cA^	1.56 ± 0.15 ^cA^	1.58 ± 0.03 ^cA^
3.0	1.66 ± 0.11 ^bA^	1.72 ± 0.13 ^bA^	1.69 ± 0.11 ^bA^
4.5	1.98 ± 0.12 ^aA^	1.95 ± 0.09 ^aA^	1.95 ± 0.09 ^aA^
Total VFA (mmol L^−1^)			
0	28.84 ± 0.52 ^cA^	29.01 ± 0.21 ^cA^	28.88 ± 0.55 ^cA^
1.5	29.31 ± 0.25 ^cA^	29.26 ± 0.15 ^cA^	29.40 ± 0.54 ^cA^
3.0	30.26 ± 0.18 ^bA^	30.41 ± 0.25 ^bA^	30.56 ± 0.21 ^bA^
4.5	31.01 ± 0.29 ^aA^	31.26 ± 0.15 ^aA^	31.24 ± 0.22 ^aA^

The results are expressed as means ± standard deviation. Capital letters ^ABC^ indicate significant differences (*p* < 0.05) in the different samples (columns). Lowercase letters ^a,b,c^ indicate significant differences (*p* < 0.05) between the same samples in different harvests.

## Data Availability

Data supporting reported results can be requested from the authors.

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
