# Peer review of "Decrease of Greenhouse Gases during an In Vitro Ruminal Digestibility Test of Forage (Festuca arundinacea) Conditioned with Selenium Nanoparticles"

_nanomaterials, 2022, doi:10.3390/nano12213823_

Round 1

Reviewer 1 Report

Dear authors,

you have done interesting experiments and obtained interesting results.

I have juse a few minor recommendations:

1. Use in vitro in italic in all manuscript, for example line 11.

2. For the beginning of the chapter Introduction, I would like to recommend to write about selenium nanoparticles and their use in agriculture. For example: Malyugina et al. Biogenic Selenium Nanoparticles in Animal Nutrition: A Review, Agriculture, 2021.

3. I recommend to use more recent references in chapters Introduction. Some of them are relatively old.

4. You have mention that "This research received no external funding." Where did you obtain financial support for this research?

I recommend publication after suggested changes.

Reviewer 2 Report

This manuscript presents the study of the evaluation about the IVDMD, VFA, the release of bioactive compounds in ruminal fluid, the TG and CH4 in an In vitro model of forage (Festuca arundinacea Schreb) enriched by foliar fertilization with Se NPs (1.5, 3.0 and 4.5 ppm) during three harvests. The results show the application of Se NPs could be a good alternative to be incorporated into forage crops that help improve digestibility parameters and reduce greenhouse gases, contributing to the commitment to reduce these gases to combat the effects of climate change. So, it can be published after minor revision.

1, Page 2, Line 76, Due to the Se NPs size can affect its uptake efficiency by plant, the characterization of exact particle size of the Se NPs is needed. If possible, I suggest the authors provide TEM image of the sample.

2, Page 5, Line 221, line 222, line 224, “600μL”, “200μL” should be “600 μL”, “200 μL”.

3, Page 7, Line 317. What caused the higher pH (8.63) in the fermentation of the forage treated with Se NPs (4.5 ppm) from the second harvest?

Reviewer 3 Report

The article "Decrease in the production of greenhouse gases with the in vitro digestibility of a forage (Festuca arundinacea) with selenium nanoparticles". The paper is in the interest of nanomaterials, however there are several issues to be solved before publication. Thus, I recommend a major revision that should cover the following points:

1. The abstract is not concise and clear enough, and needs to be improved as a whole. The data in brackets are not clearly expressed and difficult to distinguish.

2.Would you explicitly specify the novelty of your work? What progress against the most recent state-of-the-art similar studies was made?

3. The logic of the introduction part is not strong, especially the last paragraph directly throws out the research purpose of this paper without sufficient foreshadowing, and needs to be improved as a whole.

4. The Materials and Methods section needs to be rewritten, and the detection methods, experimental setup, data analysis, kinetic equations, etc. need to be reclassified. This part is too long and the classification is not clear enough. The description of the detection method is too long and too cumbersome to describe it in multiple titles. Please add the basis for the 39°C water bath. What is the basis for the design of in vitro fermentation experiments?

5. When expressing the experimental results in Section 3.1, it is necessary to describe the change key data, which corresponds to the discussion in this section, and lacks comparison with other literature data.

6. Section 3.2, please present the results of the significance analysis.

7.There is no error in Fig 1 and 2.

8. The first paragraph of Section 4 is still a description of the experimental results rather than a discussion of the experimental results and phenomena. The first sentence of the second paragraph lacks literature support. There is no comparison with other literature data in Sections 3 and 4. Whether the technology described in this paper can significantly reduce greenhouse gas emissions needs to be compared not only with the control group, but also with existing research to show the advanced nature of this research. Add mechanism analysis in Section 4, preferably in the form of pictures.

8. Please standardize the writing format. There are many formatting errors in the text, such as the first three sentences of Section 2.4, the title of Section 3.1 with an extra comma, and the conclusion with more than half brackets. The expression of data is too vague. What are the multiple data in a bracket? Which experimental group should be specified.

9. Please calculate the application cost of nanoparticles and whether it is suitable for large-scale application.

Round 2

Reviewer 3 Report

1. Heading 3.1 has an extra punctuation in front of it.

2. line285, why is p>0.05 significant, please revise carefully.
